# The Effect of J-Groove on Vortex Suppression and Energy Dissipation in a Draft Tube of Francis Turbine

Zhumei Luo [1], Cong Nie [1], Shunli Lv [1,*], Tao Guo [2,*] and Suoming Gao [3]

1   Department of Energy and Power Engineering, Kunming University of Science and Technology, Kunming 650093, China; luozhumei@163.com (Z.L.); nc1002nc@163.com (C.N.)
2   Department of Engineering Mechanics, Faculty of Civil Engineering and Mechanics, Kunming University of Science and Technology, Kunming 650500, China
3   Shenzhen Water Planning & Design Institute Company Limited, Kunming 650032, China; gaosm@swpd.cn
*   Correspondence: lvshunli@kust.edu.cn (S.L.); guotaoj@126.com (T.G.)

**Abstract:** The vortex rope in the draft tube is considered as the major contributor to pressure pulsation at partial load (PL) conditions, which causes the hydro unit to operate unstably. Based on the prototype Francis turbine HLA551-LJ-43 in the laboratory, J-grooves are designed on its conical section in this paper. We used numerical simulation to study the effect of the J-grooves on vortex suppression and energy dissipation in the draft tube. Four typical operating conditions were chosen to analyze the vortex suppression; the corresponding flow ratios $Q^*$ are 100%, 82%, 69%, and 53%, respectively. Entropy production theory is used to calculate the energy losses and assess the effect of the J-groove on energy dissipation under part-load conditions. By comparing entropy production, circumferential and axial velocity components, swirl intensity, pressure pulsation, and vortex distribution in a draft tube with and without J-grooves at different operating conditions, it can be concluded that the entropy production on the wall containing a conical section with J-grooves is obviously smaller than that without J-grooves, the effects of J-grooves on reducing circumferential velocity component $V_u$, pressure pulsation, and weakening vortex intensity and vortex rope in the conical section are obvious, especially at part load and deep part-load operating conditions. Using J-grooves shows better performance on vortex control and energy dissipation in the draft tube of a Francis turbine at partial load conditions.

**Keywords:** Francis turbine; entropy production theory; J-grooves; energy dissipation; vortex rope

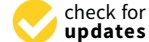



## 1. Introduction

Hydropower, the only green energy with mature technology that can be used at a large scale, has an important role in power systems, where its safe and stable operation is essential. Francis turbines often perform peak and frequency modulation tasks in the system, such that the units inevitably operate under partial load (PL) conditions and produce evident pressure pulsation when they deviate from the design flow condition.

Different degrees of vortex occur in the conical section of the draft tube, especially at 0.4 to 0.8 times the design flow, which induces pressure pulsation. Pressure pulsation can cause the output efficiency to fluctuate, resulting in the resonance of hydro units and powerhouses, which seriously affects the safety of the hydro units and powerhouses; therefore, how to suppress the pressure pulsation in the draft tube under partial load conditions effectively has become an important issue. Changing the structure of the drain cone, adding cross-shaped grilles, flow deflectors, and fins in the intake section of the draft tube, or taking measures such as air supplementation and water injection have been proposed to reduce the vibration of the draft tube [1–3]. In order to increase the operation stability of the hydro unit, researchers have been looking for a simple and easy measure to suppress the pressure pulsation in the draft tube at PL conditions.

Bosioc et al. [4] studied the flow field in the draft tube under partial load conditions with numerical simulation. A method of injecting water from the upper crown of the runner was introduced and the jet velocity was increased to 28% of the turbine flow rate, which successfully reduced the pressure fluctuation without affecting the turbine efficiency. Susan-Resiga et al. [5] also conducted experimental and numerical research on the flow feedback circuit in the vortex generator device. The pressure difference between the end of the diffuser cone and the crown of the runner was used, and a low-speed high-volume-rate jet was generated near the axis of the diffuser cone, which did not affect the overall efficiency of the turbine. Iliescu et al. [6] studied the flow field through the PIV (Particle Image Velocimetry) system under PL conditions. The speed was quantified, the diameter and position of the cavitation vortex in the draft tube model were calculated. Using PIV equipment, Goyal et al. [7] analyzed the formation of a vortex rope in a high-head Francis turbine model. Cheng [8] investigated the characteristics of vortex band, pressure pulsation, and the velocity at runner exit under PL conditions by experiments and numerical simulation. Li [9] proposed two new types of drain cones of the runner as control strategies, i.e., special-shaped drain cone and long straight drain cone, which improved the rotation characteristics of the vortex rope and optimized the pressure distribution in the draft tube through unsteady numerical simulation. Jafarzadeh [10] used the water injection method to study the effect of the rotating vortex caused by the flow instability in the draft tube on the efficiency and wear of the turbine, and the optimization of the efficiency loss during the water injection process was analyzed. Favrel [11] proposed a method for evaluating the vortex behavior and associated pressure pulsations over the entire partial load operating range and analyzed the dynamic strain on the runner.

In recent years, installing shallow grooves called "J-grooves" in some fluid machinery was proposed, which aimed to suppress various abnormal flow phenomena; however, there is very little research related to it. Kurokawa [12] studied the influence of the J-grooves on the diffuser section of the reaction turbine; the results show that the J-groove has a certain effect on the vortex control in the diffuser section. Saha et al. [13,14] proposed a kind of optimizing J-grooves to stabilize the performance curve of a mixed flow pump, which could suppress the rotating stalls in the parallel wall vanes and improve the cavitation performance for the bladeless diffuser and compressor by controlling the angular momentum of the main flow. For a bulb turbine model, Viet et al. [15] also optimized J-grooves by combining the length ratios, width ratios, angles, and numbers of grooves to suppress swirl flow in the draft tube; therefore, the use of J-grooves may be an effective method to suppress the pressure pulsation and reduce the amplitude of pressure fluctuation caused by the rotation of the vortex core around the dead water area near the draft tube inlet. However, J-grooves are mainly used in mixed flow pumps, cone diffusers of draft tubes, and other fluid machinery; currently, few studies are focused on the application in elbow draft tubes of a Francis turbine. After all, more than 80% of water turbines in the world are Francis turbines. The influence of J-grooves on the realization of vibration reduction, mechanism of vortex suppression, and energy dissipation of Francis turbines is still worth exploring.

During hydro-unit operation, the total hydraulic loss is often estimated indirectly through the efficiency formula, and it is difficult to judge the source and specific distribution of energy loss in different parts through traditional numerical simulation or experimental methods; therefore, the conventional calculation method cannot obtain accurate hydraulic loss. Nowadays, it is clear that energy dissipation should be related to entropy production. Wang [16] used the theory of entropy production to optimize the design of airfoils, centrifugal fans, and axial-flow reverse generator sets, which improved the energy utilization efficiency. Feng [17] analyzed the runaway characteristics of pumped storage units and centrifugal pumps based on entropy production theory in the simulation of the runaway process. Yu [18] numerically simulated the flow in a Francis turbine based on the SST turbulence model, the Zwart cavitation model, and the entropy production method to quantitatively calculate the irreversible energy loss, which visually shows the spatial distribution of the energy loss. In order to study the difference of the pump device in the

forward and reverse runaway transition process, Xu [19] used the fluid volume function method to analyze water–gas two-phase distribution in the upstream and downstream and combined the entropy production theory to analyze the energy loss; however, the application of entropy production theory in water turbine analysis has had less attention. In this study, the energy dissipation of the Francis turbine at PL and DPL conditions was analyzed based on entropy production theory. In addition, the effects of J-grooves on vortex control were studied by comparing circumferential and axial velocity components, swirl intensity, pressure pulsation, and vortex distribution in draft tubes with and without J-grooves at different operating conditions.

## 2. Numerical Calculation Method

### 2.1. Governing Equation

This study models the prototype turbine and performs numerical analysis based on CFD. Turbulence model RNG $k$-$\varepsilon$ that is similar to the standard $k$-$\varepsilon$ model comes from strict statistical techniques [20,21]. The turbulent kinetic energy transport equation has a wide range of applications and reasonable accuracy and has the following improvements: 1. The RNG equation adds one to $\varepsilon$ equation conditions, which effectively improves the accuracy; 2. turbulent eddies are considered to capture the vortices clearly, and the accuracy is improved [22]. The turbulence model RNG $k$-$\varepsilon$ is selected to predict the performance of the water turbine. The transportation equations of the RNG $k$-$\varepsilon$ model are

$$\frac{\partial}{\partial t}(pk) + \frac{\partial}{\partial x_i}(pku_i) = \frac{\partial}{\partial x_j}\left(a_k\mu_{eff}\frac{\partial k}{\partial x_j}\right) + G_k + G_b - \rho\varepsilon - Y_M + S_k \tag{1}$$

$$\frac{\partial}{\partial t}(\rho\varepsilon) + \frac{\partial}{\partial x_i}(\rho\varepsilon u_i) = \frac{\partial}{\partial x_j}\left(\alpha_\varepsilon\mu_{eff}\frac{\partial\varepsilon}{\partial x_j}\right) + C_{1\varepsilon}\frac{\varepsilon}{k}(G_k + C_{3c}G_b) - C_{2\varepsilon}\rho\frac{\varepsilon^2}{k} - R_\varepsilon + S_\varepsilon \tag{2}$$

where $k$ and $\varepsilon$ represent the turbulent kinetic energy and dissipation rate, respectively, $G_k$ represents the generation of turbulent kinetic energy caused by the average velocity gradient, $G_b$ is the turbulent kinetic energy generated by buoyancy, and $Y_M$ represents the contribution of fluctuating expansion in compressible turbulence to the total dissipation rate, $\mu_{eff}$ is the dynamic viscosity coefficient of flow, $C_{1\varepsilon}$, $C_{2\varepsilon}$, and $C_{3\varepsilon}$ are all constants. $S_k$ and $S_\varepsilon$ are user-defined source terms, and $R_\varepsilon$ is an additional term given by the equation. Based on the scale elimination process in RNG theory, the differential equation of turbulent viscosity is derived as follows:

$$d\left(\frac{\rho^2 k}{\sqrt{\varepsilon u}}\right) = 1.72\frac{v}{\sqrt{v^3 - 1 + C_V}}dv \tag{3}$$

where $v$ represents the scale change of low Reynolds number in turbulent transmission, $v = \mu_{eff}/\mu$, $C_V$ is the constant of fluctuation caused by fluid diffusion, $C_V \approx 100$. The above equations are integrated to obtain an accurate description of how effective turbulent transport varies with the effective Reynolds number (or eddy current standard), allowing the model to handle near-wall flow efficiently [23,24].

### 2.2. Boundary Conditions

In order to obtain the pressure fluctuation and vortex rope in the draft tube of the Francis turbine and evaluate the suppression effect of J-grooves on the vortex, a transient CFD analysis is conducted. The entire fluid domain of a hydraulic turbine is formed by combining the interfaces between the spiral casing and the guide vanes, the guide vanes and the runner, and the runner and the draft tube. In detail, handling data exchange between rotating part runner and stationary part by sliding mesh technology to handle the moving boundaries. The entrance of the spiral casing is set as the velocity inlet, and the outlet of the draft tube is taken as the static pressure outlet, one standard atmospheric pressure. All solid surfaces of the turbine are provided with nonslip wall boundary conditions with a

wall roughness of 0.5 mm. The liquid phase is clean water; the density is $\rho = 997 \text{ kg/m}^3$. We use the results of the steady analysis as the initial conditions for the transient analysis. The time step is the time that it takes the runner to spin 3 degrees. The convergence criteria require that the residuals should be less than $10^{-4}$.

### 2.3. Entropy Production Theory

When the turbine is operating at part load, the backflow phenomenon, pressure pulsation, vortex rope, flow separation in the draft tube, etc., will increase the energy loss [25]. The location of high hydraulic losses cannot be accurately determined using traditional differential pressure assessment methods. The local entropy production rate (LEPR) is closer to the physical quantity of flow loss than the head loss coefficient calculated by the differential pressure method. According to the second law of thermodynamics, a small increment of entropy is always greater than 0; therefore, the entropy production theory can be used to calculate the energy losses and assesses the effect of the J-groove on energy dissipation under PL condition, which provides a method to calculate the operation efficiency of a water turbine accurately.

Entropy production due to viscous dissipation can be directly calculated [26]. For an incompressible flow, the viscous dissipation function $\phi$ is expressed.

$$\phi = 2\mu_{eff}\left[\left(\frac{\partial u_1}{\partial x_1}\right)^2 + \left(\frac{\partial u_2}{\partial x_2}\right)^2 + \left(\frac{\partial u_3}{\partial x_3}\right)^2\right] + \mu_{eff}\left[\left(\frac{\partial u_1}{\partial x_2} + \frac{\partial u_2}{\partial x_1}\right)^2 + \left(\frac{\partial u_1}{\partial x_3} + \frac{\partial u_3}{\partial x_1}\right)^2 + \left(\frac{\partial u_2}{\partial x_3} + \frac{\partial u_3}{\partial x_2}\right)^2\right] \tag{4}$$

when the specific volume of water is large, the temperature of the water remains unchanged when the water turbine is operating, so the entropy production caused by heat transfer is not considered. In turbulent flow, the local entropy generation rate (LEPR) $\dot{S}_D'''$ based on Reynold-time-averaged motion consists of two terms. One is the specific entropy production rate caused by time-averaged movement, called direct entropy production rate $\dot{S}_{\overline{D}}'''$; the other is the specific entropy production rate caused by velocity fluctuation, called turbulent entropy production rate $\dot{S}_{D'}'''$; therefore, the total entropy production rate is expressed as

$$\dot{S}_D''' = \dot{S}_{\overline{D}}''' + \dot{S}_{D'}''' \tag{5}$$

where

$$\dot{S}_{\overline{D}}''' = \frac{2\mu_{eff}}{T}\left[\left(\frac{\partial \overline{u}_1}{\partial x_1}\right)^2 + \left(\frac{\partial \overline{u}_2}{\partial x_2}\right)^2 + \left(\frac{\partial \overline{u}_3}{\partial x_3}\right)^2\right] + \frac{\mu_{eff}}{T}\left[\left(\frac{\partial \overline{u}_1}{\partial x_2} + \frac{\partial \overline{u}_2}{\partial x_1}\right)^2 + \left(\frac{\partial \overline{u}_1}{\partial x_3} + \frac{\partial \overline{u}_3}{\partial x_1}\right)^2 + \left(\frac{\partial \overline{u}_2}{\partial x_3} + \frac{\partial \overline{u}_3}{\partial x_2}\right)^2\right] \tag{6}$$

$$\dot{S}_{D'}''' = \frac{2\mu_{eff}}{T}\left[\left(\frac{\partial u_1'}{\partial x_1}\right)^2 + \left(\frac{\partial u_2'}{\partial x_2}\right)^2 + \left(\frac{\partial u_3'}{\partial x_3}\right)^2\right] + \frac{\mu_{eff}}{T}\left[\left(\frac{\partial u_1'}{\partial x_2} + \frac{\partial u_2'}{\partial x_1}\right)^2 + \left(\frac{\partial u_1'}{\partial x_3} + \frac{\partial u_3'}{\partial x_1}\right)^2 + \left(\frac{\partial u_2'}{\partial x_3} + \frac{\partial u_3'}{\partial x_2}\right)^2\right] \tag{7}$$

Among them, $T$ represents temperature; $u_i$ and $u_i'$ are time-averaged velocity and turbulent fluctuation velocity, respectively; $i$ is the three directions in the Cartesian coordinate system ($i$ = 1, 2, 3); $\mu_{eff}$ can be calculated through the following formula

$$\mu_{eff} = \mu + \mu_t \tag{8}$$

where $\mu$ is the laminar dynamic viscosity; $\mu_t$ is the turbulent dynamic viscosity. The local entropy production rate due to velocity fluctuations can be approximately calculated as follows:

$$\dot{S}_{D'}''' = \beta\frac{\rho\omega k}{T} \tag{9}$$

In the above formula, the empirical coefficient $\beta$ is obtained by direct numerical simulation, $\beta$ = 0.09 (additional literature); $\omega$ represents the turbulent eddy–viscous frequency having units $\text{s}^{-1}$.

## 3. Numerical Calculation Method

### 3.1. Prototype Model and Mesh Generation

The prototype Francis turbine HLA551-LJ-43 in Fluid–Solid Coupling Laboratory is shown in Figure 1. The turbine contains a spiral case whose entrance diameter is 0.55 m, a runner with 13 blades, 16 stay vanes, 8 guide vanes, and an elbow draft tube. The rated speed of runner $n_r$ = 600 rpm, under the specification of design head and discharge, the turbine has an output of 55 kW when efficiency is 92%. The turbine parameters are shown in Table 1.

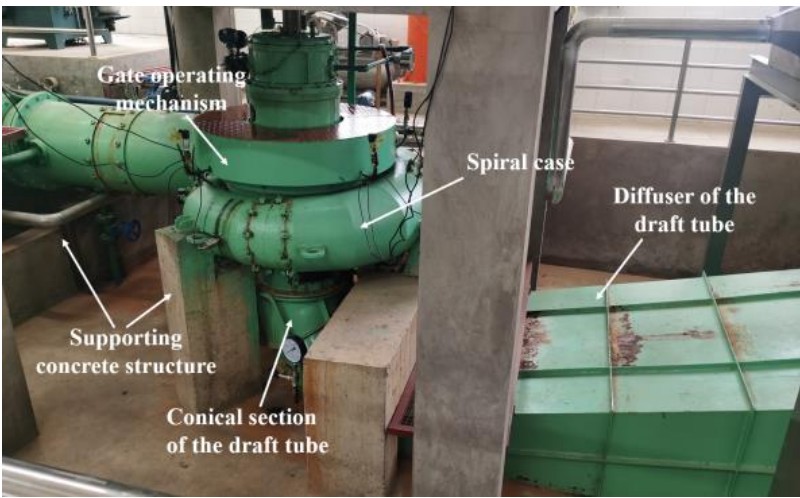

**Figure 1.** Prototype hydro turbine HLA551-LJ-43.

**Table 1.** The parameters of the HLA551-LJ-43.

| Parameter | Symbol | Value |
|---|---|---|
| Nominal diameter of runner | $D_1$ | 0.43 m |
| Rated unit speed | $n_{11}$ | 81.587 r/min |
| Rated unit flow | $Q_{11}$ | 1.064 m$^3$/s |
| Number of runner blades | $Z_r$ | 13 |
| Stay vanes number | $Z_T$ | 16 |
| Guide vanes number | $Z_s$ | 8 |
| Entrance diameter of spiral casing | $D_0$ | 0.547 m |

SolidWorks modeling software was used to model and mesh the prototype turbine HLA551-LJ-43. The complete three-dimensional flow passage of the model turbine for this simulation is shown in Figure 2, including the spiral casing, stay vanes, guide vanes, runners, and draft tube. Out of these components, the runner is a rotating part, and the others are stationary; therefore, each component of the turbine has been modeled separately in Ansys Workbench and then assembled through suitable interfaces so that the turbine can be simulated as a whole structure.

In this paper, the ICEM software is used to divide the entire computational domain into structured hexahedral meshes. To ensure calculation accuracy and efficiency, grid independence verification is performed on the entire hydraulic turbine, the efficiency of the turbine η at full load is evaluated by changing the number of nodes. The mesh numbers of the spiral casing, guide vanes, runner, and draft tube are 3.8 million, 2.1 million, 4.74 million, and 2.32 million, respectively (as seen in Figure 3). As shown in Figure 4, the η can reach about 92% and does not change significantly when the total grid cells are above 13 million. Under the condition that the flow parameters remain unchanged, the denser the boundary layer grid, the smaller the value of y+. Due to the small size of the turbine in this

experiment, the grid has reached 13 million, the boundary layer is densified, the grid is fine, and check the y+ value by software to be 1.

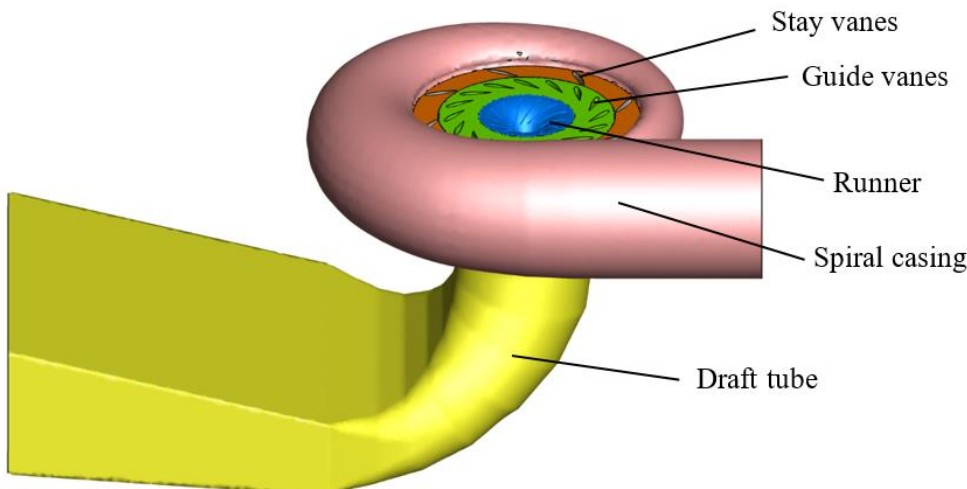

**Figure 2.** Three-dimensional perspective view of the turbine model.

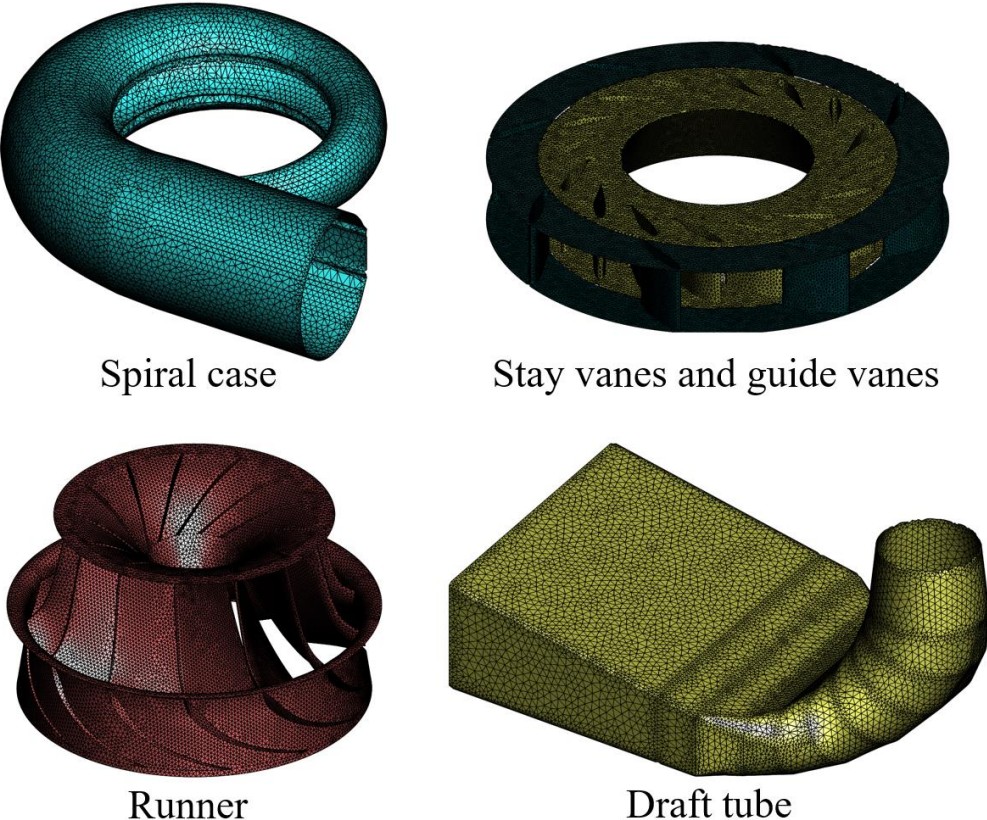

**Figure 3.** Structured hexahedral grids of each flow component.

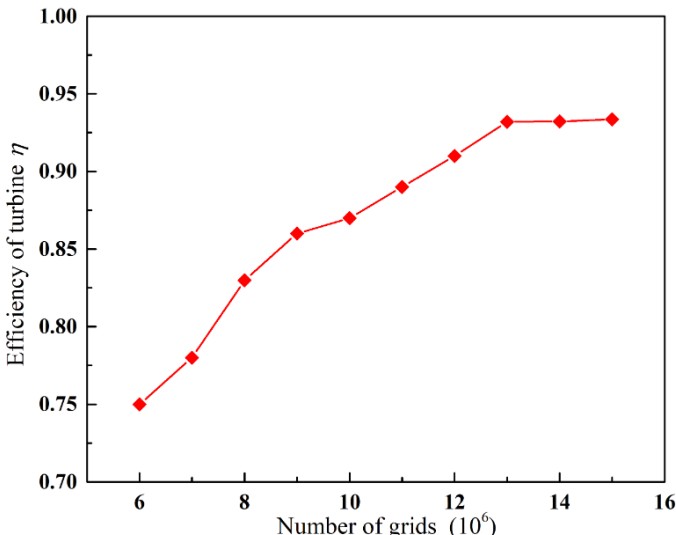

**Figure 4.** Grid independence verification.

### 3.2. Application of J-Grooves in Draft Tube of Francis Turbine

The shape of the draft tube is an important factor in maintaining stable flow conditions and suppressing eddy currents in this domain. From the research of the previous scholars, J-grooves are a kind of structure modification applied in some fluid machinery that is beneficial to minimize the vibration of the fluid machinery. By controlling the angular momentum of the main fluid flow, J-grooves contribute to suppressing the anomalous flow phenomena. If the grooves are installed parallel to the pressure gradient on the outer wall of the draft tube, the pressure gradient of the main water flow in the draft tube will cause a strong reverse flow. When the swirling flow enters the groove, the main flow loses angular momentum. The fluid from the groove mixes with the main water flow, thereby reducing the angular momentum of the main flow.

In this paper, the effect of vortex suppression and energy dissipation of J-shaped grooves is studied by the combination of draft tubes and J-grooves. As shown in Figure 5, the J-grooves on the interface of the straight cone section is evenly distributed along the circumferential direction. According to the optimal layout of J-grooves in Ref. [15], 12 J-grooves are the most suitable number for the turbine HLA551-LJ-43, and the length $L$ is 120 mm, the width of each J-groove $W$ is 10 mm. $D_3$ is the inlet diameter of the draft tube. In order to analyze the changes in pressure and vortex intensity before and after the water flow enters the J-grooves, two monitoring points TS1, TS2, and one monitoring surface TP1, were set up on the draft tube; TP1 is just the exit surface of the conical section.

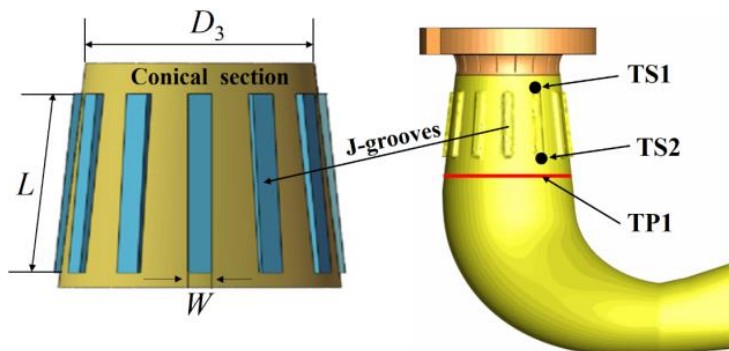

**Figure 5.** The layout of the J-grooves and the monitoring points and surface in the draft tube.

### 3.3. Experimental Operating Point

The comprehensive characteristic curve of the model and the positions under four typical working conditions are shown in Figure 6. The abscissa $Q_{11}$ and ordinate $n_{11}$ represent the unit flow rate and the unit speed, respectively. Four typical operating points were chosen: the best efficiency point BEP, the other three partial load points NV, PL, and DPL under the design head. The flow ratios $Q^*$ ($Q^* = Q/Q_{BEP}$, $Q_{BEP}$ is the flow at the maximum efficiency) of the four working conditions BEP, NV, PL, DPL are 100%, 82%, 69%, and 53%, respectively. The best efficiency point corresponds to 93.2% of operation efficiency, as shown by the orange dots in Figure 6. The guide vane opening is 22.6 mm, unit speed $n_{11}$ is 79.9 r/min, and unit flow $Q_{11}$ is 1.064 m$^3$/s.

When the hydro turbine is running, the vortex rope in the draft tube shows different forms under different working conditions. The optimization effects of the J-groove on the unstable operation and reduced efficiency of the turbine caused by pressure fluctuation, swirl intensity, and energy loss are discussed at the four typical operating points, which are to change the axial velocity and angular momentum of water flow in the draft tube and explore the mechanism of vibration caused by vortex rope. In this paper, the research on the eddy current changes and flow characteristics at different operation conditions with or without J-grooves was carried out.

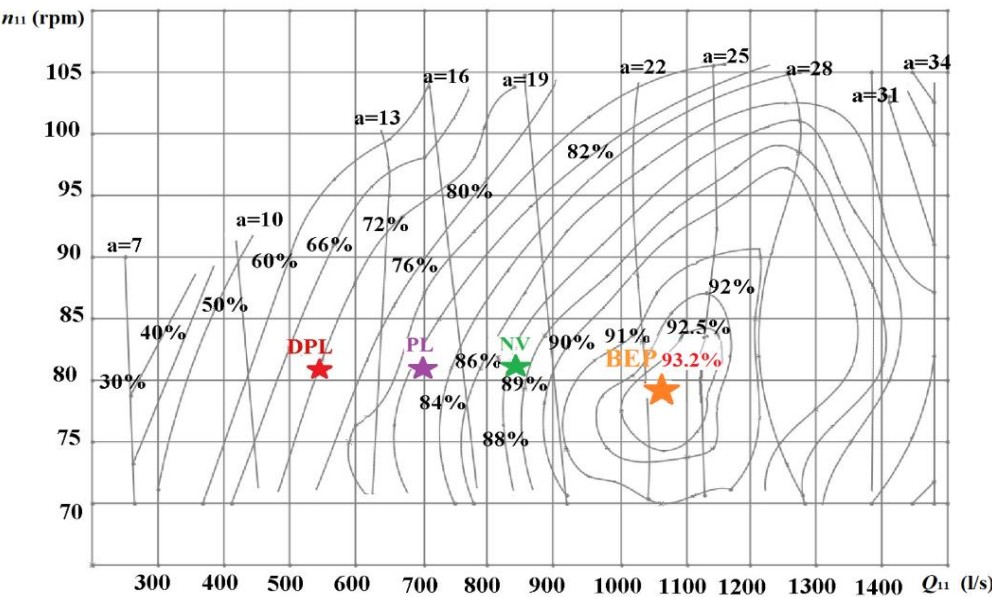

**Figure 6.** Comprehensive characteristic curve of HLA551 and its four experimental points.

## 4. Results and Discussions

### 4.1. Influence of J-Grooves on Energy Dissipation Based on Entropy Production Theory

Compared with the traditional analysis approach of pressure and velocity fields, the entropy production method is more intuitive to analyze energy dissipation, which helps to better know the accurate location where the hydraulic losses occur and understand the complex flow. When the unit operates at DPL operating condition ($Q^* = 53\%$), the peripheral speed at the runner outlet is high, resulting in a higher vibration of the hydro unit and energy losses. The $Q^* = 53\%$ operating condition was chosen to evaluate the effect of the J-grooves on reducing the energy losses of the unit and analyze the performance of the turbine before and after installed J-grooves through entropy production theory. The energy loss was quantitatively analyzed, and the spatial distribution of the energy loss at DPL condition is visually displayed, as shown in Figure 7. To quantitatively observe the energy loss of the runner and the straight conical section of the draft tube, the LEPR distribution on two monitoring surfaces was taken at the lower edge of the turbine runner

blade and the conical section of the draft tube, namely Sections 1 and 2. In this paper, N-J represents the water turbine without J-grooves; W-J means the draft tube installed J-grooves on its wall of the conical section.

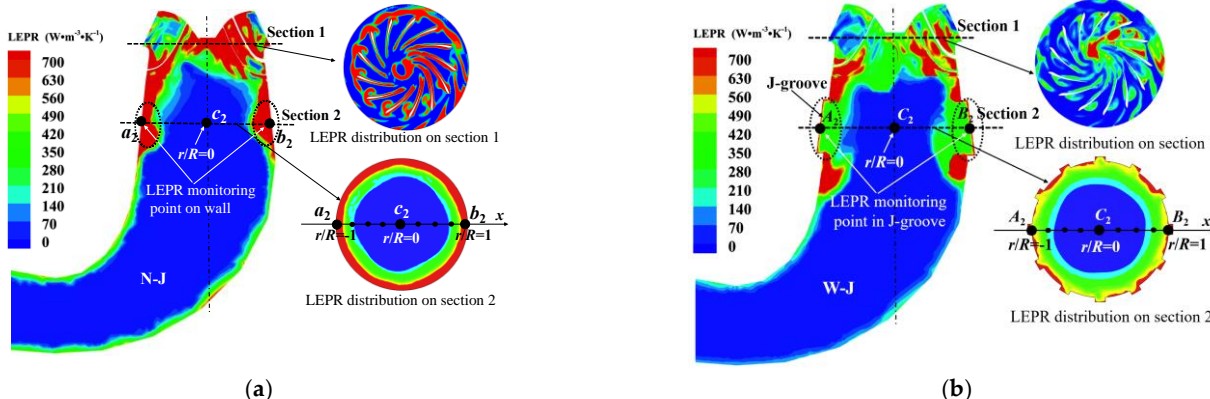

**Figure 7.** LEPR distribution in runner and draft tube with or without J-groove when $Q^* = 53\%$. (**a**) The entropy production rates without J-groove, (**b**) The entropy production rates with J-grooves.

Taking the center of Section 2 as the origin point, $R$ is the radius of Section 2, and $r$ is the position from the origin point in the $x$-direction. It can be seen from Figure 7a,b, under the DPL condition, the water flow at Section 1 has a larger positive angle of attack on the blade, and the circumferential component at the runner outlet is larger, resulting in a larger hydraulic loss; therefore, it can be observed that there is a strong high LEPR at the edge of the runner outlet, and the local entropy production rates close to the runner blades are larger. After the J-groove is installed, the direction of water flow at the outlet of the runner is suppressed by the J-groove. Under the action of the backflow in the conical section, LEPR is reduced but distributed unevenly.

It can be clearly seen that in Section 2, the high LEPR is distributed in the regions of $r/R = -1\sim-0.5$ and $0.5\sim1$ ($r/R$ represents relative radius, $r/R = 0$ is the center of Section 2) from the wall to the center, and gradually decrease along the wall to the center. The water flow in this area is closely attached to the draft tube wall with its own centrifugal force, resulting in a significant increase in entropy production. After the J-groove is installed, the high entropy production only existed in the area close to the wall surface in the groove, near points $A_2$ and $B_2$ in Figure 7b, which is significantly smaller than the entropy production at the walls $a_2$ and $b_2$ without J-grooves. Because of the suppression of the J-grooves to the circumferential velocity component of the water flow at the runner outlet, part of the water flow gradually changes from radial to axial direction along the circulating flow direction. In addition, the area of high-entropy production near the wall is significantly reduced, the hydraulic loss is also reduced.

Figure 8 directly shows the LEPR in the $r/R$ direction at nine monitoring points on Section 2 (see in Figure 7) when the $Q^*$ are 53% (N−J), 53% (W−J), 69% (N−J), and 69% (W−J), respectively. The spacing ratio between two adjacent monitoring points is 0.25. It can be observed that the variation trends of the LEPR in the $r/R$ direction are basically similar in the above four cases. In the range of $r/R = -0.5\sim0.5$, the entropy productions are low and stable. While for the case $r/R = -1\sim-0.5$ and $0.5\sim1$, the entropy productions variate and decrease from the wall to the center of Section 2 gradually. This result is consistent with the visualized map of entropy production distribution in Figure 7. No matter at DPL or PL condition, the entropy production on the wall of the conical section with J-grooves is obviously smaller (more than 100 W·m$^{-3}$·K$^{-1}$) than that without J-grooves, it is mainly because the peripheral velocity component at the outlet of the runner is suppressed after installing J-grooves, so that part of the flow circulating at the near-wall by centrifugal force flows along the axial direction, reducing diffusion cycle near the wall. Consequently, the

entropy production near the J-groove is reduced, and J-grooves have an obvious effect on reducing energy loss in the draft tube of a Francis turbine.

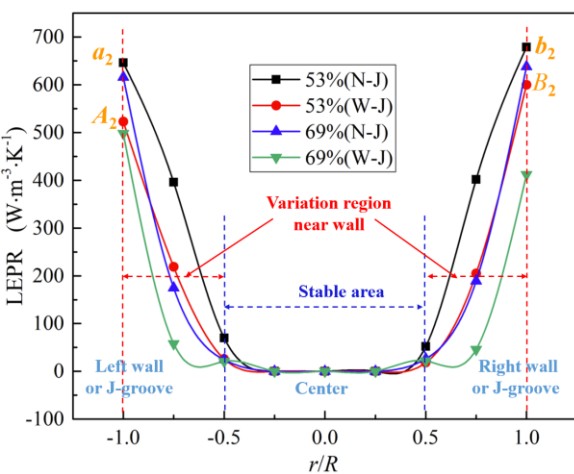

**Figure 8.** LEPR on Section 2 of the draft tube at DPL and PL conditions with or without J-groove.

### 4.2. Influence of J-Grooves on Velocity Distribution

At partial loads, the water no longer flows along the so-called "structural line" of the runner after entering the runner, but rapidly deflects downward the band under the centrifugal force [27]. As a result, the rotating speed of the water flow in the runner blade channel increases, and the water flow entering the draft tube inevitably has a certain circumferential speed $V_u$, leading to a vacuum appearing in the center of the conical section. This phenomenon creates the conditions for backflow and recirculation in the draft tube and prepares the final conditions for the generation of vortex rope and vortex pressure fluctuation.

It is well known that the circumferential speed $V_u$ is the unique speed of producing vortex under partial load working conditions; axial speed $V_z$ is inherent [28]. Under the four typical operating conditions, the distributions of $V_u$ and $V_z$ along the relative radius $r/R$ are calculated in the conical section with and without J-grooves, as shown in Figure 9. The purpose is to obtain the effect of the J-groove on improving the velocity distribution and vortex suppression.

In Figure 9, the circumferential and axial velocity components $V_u$ and $V_z$ of the flow are unevenly distributed on the cross section of the draft tube and change with different working conditions. On the cross section of the conical section, $V_u$ gradually decreases from the tube wall to the center at DPL, PL, NV, and BEP operating conditions.

When installed the J-grooves, the countercurrent in the central area increases with the increase in the axial velocity near the draft tube wall compared with the case without J-grooves. When the flow on the wall is mixed with the main flow, the axial velocity in the central area is almost the same as that without J-grooves. Figure 9a,b show that the value of $V_u$ in the straight cone is lower than that when the J-grooves are not installed. The water flow in the draft tube without J-grooves has obvious eccentricity, especially under low load conditions. For the draft tube installed J-grooves, the circumferential velocity $V_u$ is reduced by approximately 0.7 m/s compared with the draft tube without J-grooves at $r/R = -0.25$. That means J-grooves can reduce the circumferential velocity component $V_u$ effectively. The rotation speed of the vortex core is decreased, thereby the pressure level of the vortex core is increased, which constrains the eccentricity of the vortex rope, the pressure fluctuation is decreased consequently. On the contrary, the axial speeds $V_z$ do not change insignificantly at DPL and PL conditions for the draft tube with or without J-grooves.

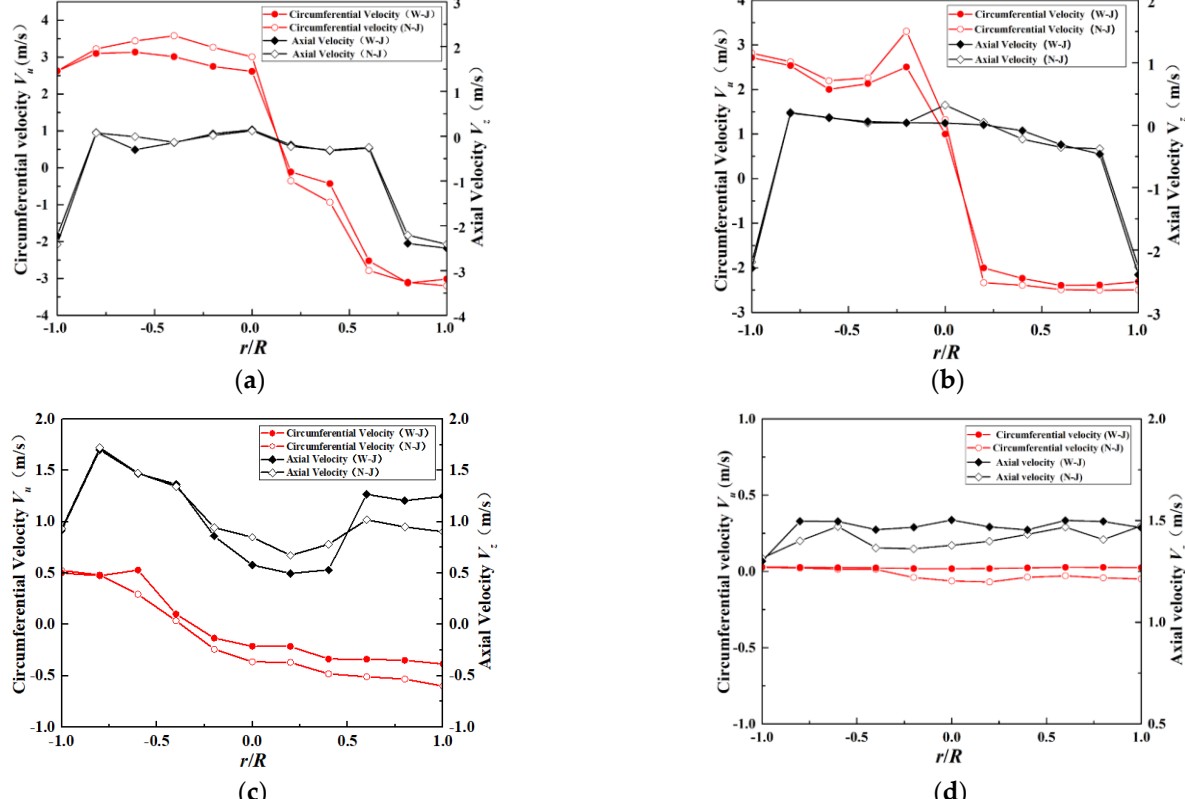

**Figure 9.** The distributions of circumferential and axial velocity in conical section with or without J-grooves. (**a**) $Q^* = 53\%$, (**b**) $Q^* = 69\%$, (**c**) $Q^* = 82\%$, (**d**) $Q^* = 100\%$.

When hydro units operate in better conditions, as shown in Figure 9c,d, pressure fluctuations are not obvious, especially when $Q^* = 100\%$, the $V_u$ is close to 0. The direction of the absolute velocity of the flow out from the runner blades is normal with no vortex and shedding. Consequently, the flow field is relatively stable. For the draft tube with and without J-grooves, the $V_u$ is almost the same, while the axial speed does not change much in the entire cross section.

### 4.3. Influence of J-Grooves on Swirl Intensity

Swirl intensity, also known as vortex intensity, is used to evaluate the vortex characteristics of the draft tube with or without J-grooves under different vortex conditions. It is the ratio of the axial flux of angular momentum to the axial flux of axial momentum [29]. Swirl intensity is expressed by $S_{num}$, as shown in Equation (10).

$$S_{num} = \frac{\int_r V_z V_u r^2 dr}{R \int_r V_z^2 r dr} \tag{10}$$

where $V_z$, $V_u$, $r$, and $R$ represent the velocity components in the axial and circumferential directions, the radial position, and the radius of the draft tube cross section, respectively. In order to compare the vortex intensity of the draft tube with and without J-grooves, the $S_{num}$ at DPL, PL, NV, and BEP operating conditions are calculated, as shown in Figure 10. $D_{TS2}$ is the diameter of the plane where the monitoring point TS2 is located.

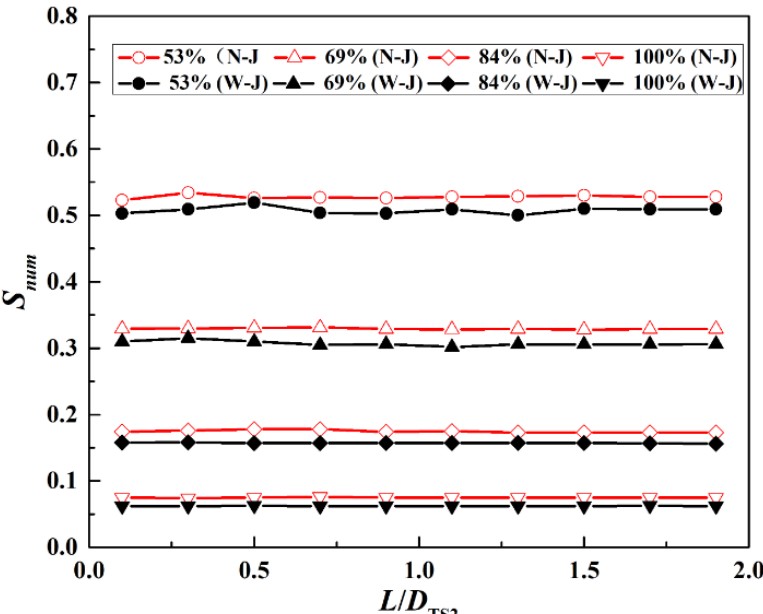

**Figure 10.** The swirl intensity $S_{num}$ under four operating conditions.

At the exit of the runner, the absolute angle of the flow is much less than 90° under partial load, so a large circumferential velocity component $V_u$ existed when flow enters to the conical section [30]. It can be seen from Figure 10 that at DPL and PL operating conditions ($Q^* = 53\%$, $Q^* = 69\%$), the $S_{num}$ is significantly higher than that of the other two working conditions, NV and BEP. $S_{num}$ decreases with the increases in load value, no vortex in the flow occurs in the groove and $S_{num}$ is almost 0 at the full load ($Q^* = 100\%$). It is clear to find that under the same operating conditions, swirl intensity in the draft tube with J-grooves is lower than that without J-grooves. The reduction in eddy current contributes to the increase in axial momentum and the decrease in angular momentum [31]. In addition, the axial momentum increases sharply in the conical section, while the angular momentum suddenly decreases at the end of the J-grooves; therefore, the draft tube installed with J-groove reduces the eddy current and increases the axial speed to a certain extent. The results in Figure 10 prove the significant role of the J-groove on vortex suppression.

In order to evaluate the influence of J-groove on vortex suppression quantitatively, the following reduction coefficient $C_r$ is used:

$$C_r = 1 - \frac{(S_{out}/S_{num})_{\text{W}-\text{J}}}{(S_{out}/S_{num})_{\text{N}-\text{J}}} \tag{11}$$

In the formula, $S_{out}$ is the swirl intensity at the outlet of the draft tube. When $C_r = 1$, the swirls are completely suppressed.

Figure 11 shows the relationship between $S_{num}$ and reduction coefficient $C_r$. It can be seen from Figure 11, as the load decreases, the vortex intensity $S_{num}$ increases under the action of a larger circumferential speed $V_u$ at the outlet of the runner. After installing the J-grooves, the angular momentum is effectively reduced, and the swirl intensity is weakened. Consequently, when the $S_{num}$ is reduced, $C_r$ becomes larger as the load decreases, and the effect of J-grooves on vortex suppression is more evident. The weakening of the vortex force makes the energy dissipation effect obvious, and the efficiency of the draft tube is improved [32]. It proves that the installation of the J-groove in the conical section exerts a suppressive effect on swirl intensity.

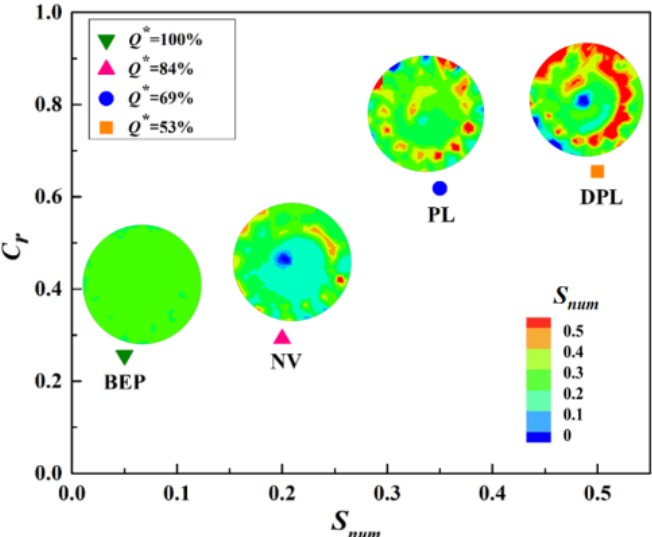

**Figure 11.** The relationship between the swirl intensity $S_{num}$ and the reduction coefficient $C_r$.

### 4.4. Influence of J-Grooves on the Pressure Fluctuation

In the critical partial load range, the rotating pressure field of the vortex and the pressure field of the main flow are superimposed to form an eccentric rotating pressure in the draft tube [33]. As a result, the pressure pulsations generate on the wall of the draft tube, especially in the straight cone and elbow sections. The closer the lowest pressure point is to the pipe wall, the greater the pressure pulsation amplitude [34,35].

Three basic parameters of pressure fluctuation are amplitude, frequency, and phase. Among the three parameters, a phase has no actual effect on the general mechanical vibration or hydraulic vibration. Accordingly, the amplitude and frequency of pressure fluctuation are studied [36].

In order to understand the influence of J-grooves on the pressure fluctuation of the vortex rope in detail, the pressures of the draft tube with or without J-grooves on the monitoring surface TP1 were investigated, as shown in Figure 12. Through unsteady calculation, the pressure change with time on the monitoring surface TP1 of the draft tube was obtained. Whether there is a J-groove or not, a spiral vortex core around the dead water area appears. Because two partial load conditions DPL ($Q^* = 53\%$) and PL ($Q^* = 69\%$) are prone to generate vortex ropes during operation, only the pressure cloud diagram under the two conditions was studied. It can be clearly found that no matter the DPL ($Q^* = 53\%$) condition or PL ($Q^* = 69\%$) condition, eccentric vortex ropes are generated. As shown in Figure 12a,b, at flow ratio $Q^* = 53\%$ working condition, the pressure at the center of the vortex increases from $-3000$ Pa without J-grooves to 4000 Pa after installing J-grooves. Due to the pressure in the center of the vortex increasing significantly, the vacuum value in the vortex core zone decreases, which is a benefit to the improvement of the cavitation phenomenon in the draft tube. The same results also appear in $Q^* = 69\%$ operating condition (Figure 12c,d). It shows that the J-grooves can effectively improve the cavity vortex rope of the draft tube and reduce the probability of cavity cavitation.

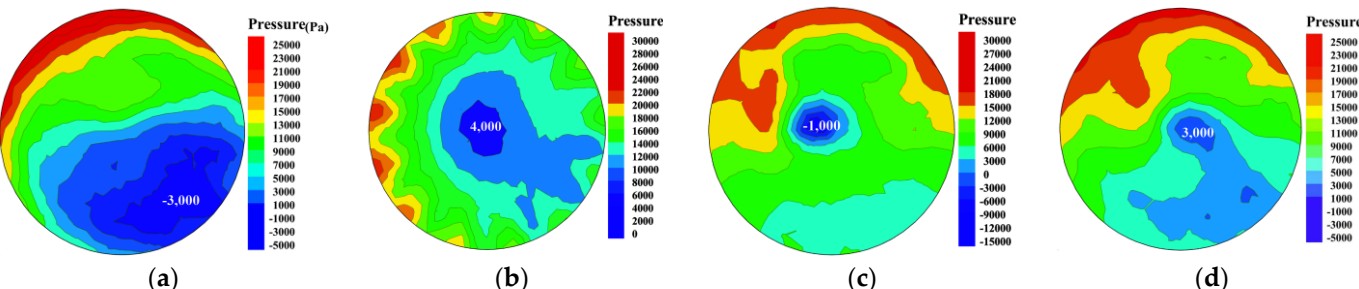

**Figure 12.** Pressure contours on TP1 surface of draft tube. (**a**) $Q^*$ = 53%(N−J), (**b**) $Q^*$ = 69% (W−J), (**c**) $Q^*$ = 82%(N−J), (**d**) $Q^*$ = 100% (W−J).

Periodic pressure pulsation is found at the two monitoring points TS1 and TS2, in the straight cone section. Figure 13 illustrates that the pressure changes with time at the two monitoring points TS1 and TS2, under the four operating conditions. Whether the draft tube with J-grooves or without J-grooves, the amplitudes of pressure pulsation at TS1 point are apparently greater than that of TS2 point and both have large amplitudes at DPL and PL conditions. When the hydro unit runs near the rated output, the amplitudes of pressure pulsation are significantly weakened, which indicates that there is no spiral vortex rope occurred.

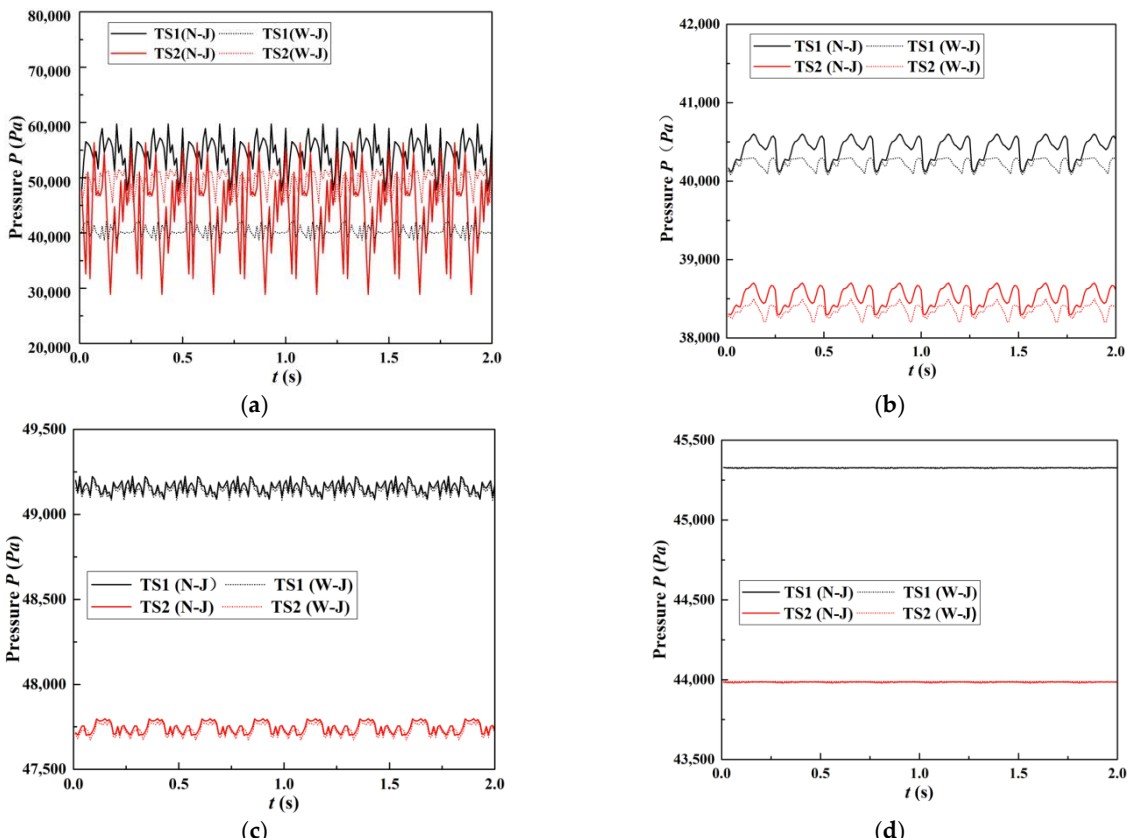

**Figure 13.** Pressure fluctuation at monitoring points TS1 and TS2 under four operating conditions. (**a**) $Q^*$ = 53%, (**b**) $Q^*$ = 69%, (**c**) $Q^*$ = 84%, (**d**) $Q^*$ = 100%.

The pressure pulsation in the draft tube is caused by the appearance of the spiral vortex. Because the vortex core is located at the eccentric position of the draft tube cross section, when the water flows through the J-grooves, periodic rotation of non-uniform flow velocity, pressure field, the asymmetry of the velocity, and pressure distribution on the cross section are all improved [37]; therefore, the periodic pressure pulsations generated

on the draft tube wall are weakened. As shown in Figure 13a,b, under $Q^* = 69\%$ load condition, the pressure fluctuations at the monitoring points also are evident, but they are not as severe as those when $Q^* = 53\%$. Compared with no J-grooves, amplitudes of pressure pulsation at TS1 and TS2 points have a significant decreasing trend in the case of J-grooves installation, especially in part-load conditions. The pressure fluctuations are evidently suppressed after the J-grooves are installed. In Figure 13c,d, the pressure pulsations are no longer obvious at NV and BEP working conditions; the turbine runs smoothly without obvious vortex rope.

Through unsteady calculation, the pressure change with time at monitoring points TS1 and TS2 were obtained. Under four operating conditions, the spectrum signals of pressure fluctuation at monitoring points TS1 and TS2 with or without J-grooves were studied to judge the suppression effect of the J-groove on pressure fluctuation. As shown in Figure 14, the complex low-frequency components, main frequency, and secondary frequency can be observed. After installing J-grooves, the spectrum peak of the main frequency and the harmonic component are significantly reduced, and the pressure pulsation phenomenon is weakened. The spectrum peak of the pressure fluctuation is very high at the DPL condition; it is far greater than the value under the BEP working condition, which indicates that the pressure fluctuation is extremely obvious when the unit operates at part-load condition.

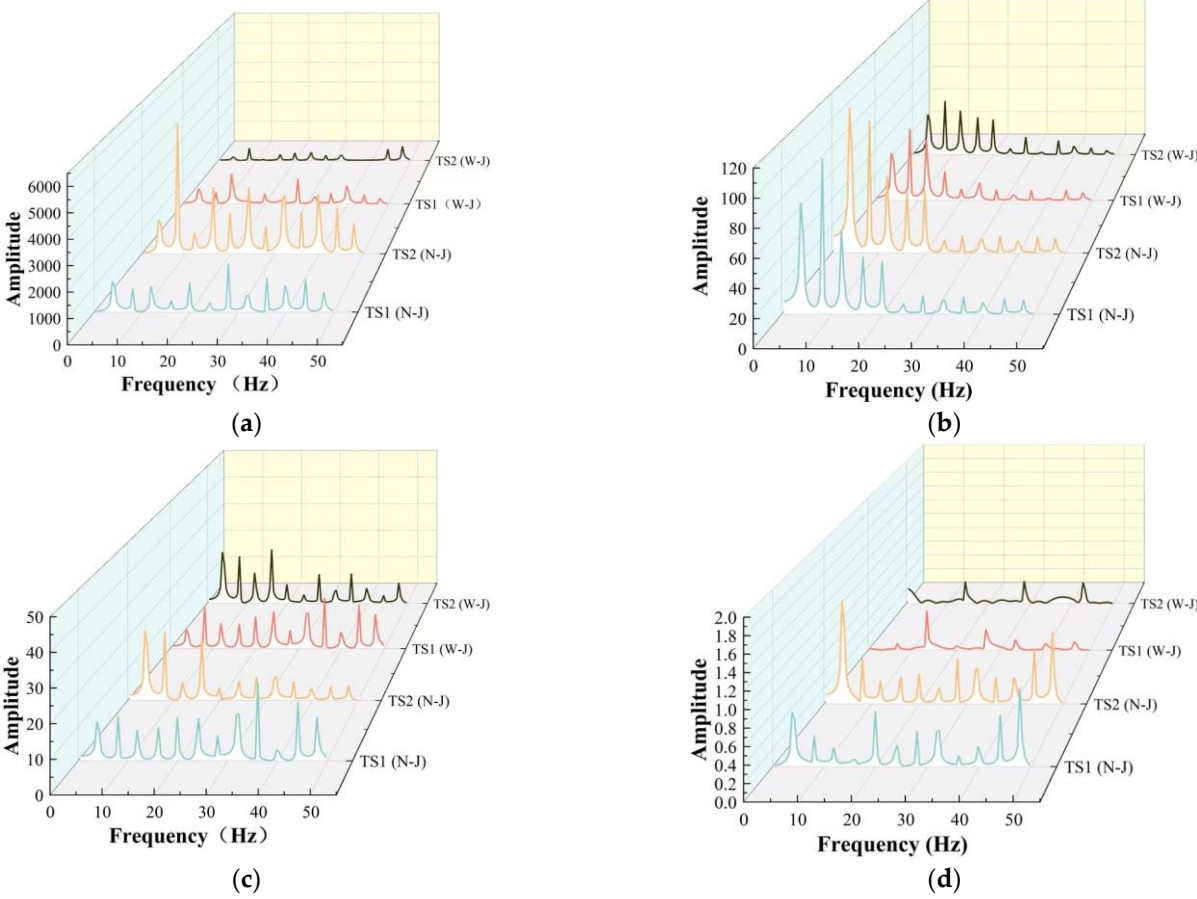

**Figure 14.** Pressure pulsation spectrum at monitoring points TS1 and TS2 under four operating conditions. (**a**) $Q^* = 53\%$, (**b**) $Q^* = 69\%$, (**c**) $Q^* = 84\%$, (**d**) $Q^* = 100\%$.

Take DPL ($Q^* = 53\%$) condition as an example to analyze the effect of the J-grooves on improving the pressure pulsation of the vortex rope, as shown in Figure 14a. Without J-grooves, the frequency of the pressure pulsation is 27 Hz at TS1 monitoring point, which is about 1/3.6 of the rotation frequency of the turbine and is consistent with the statistical results. Moreover, the peak value of the spectrum reaches 2000. After setting J-grooves,

the pressure pulsation at the TS1 monitoring point still exists, and its main frequency and harmonic components are basically the same as those without J-grooves; however, the spectrum peak of the main frequency has dropped to about 1000, which is significantly reduced. Compared with the spectrum peaks of TS2 (N-J) and TS2 (W-J), it can be found that the peak of the pressure pulsation spectrum is significantly weakened after the water flows through J-grooves. When $Q^* = 84\%$ and $Q^* = 100\%$, units run stably, the low-pressure area and high-pressure area tend to be the same, the amplitudes of pressure fluctuations do not change much, and the positive suppression effect of the J-grooves can also be observed, but it is not as obvious as the results when the unit operates at DPL operating condition.

### 4.5. Influence of J-Grooves on Vortex Distribution

The rotational nature of motion, the diffusivity of eddies, and the dissipation of energy are the basic properties of hydraulic turbine fluids. The direct reason for the vortex formation is the vortex flow at the exit of the runner, which is closely related to the evolution of the vortex [38]. In order to identify the vortex suppression and energy dissipation effect of the J-grooves on the draft tube in the Francis turbine, the vorticity transport equation is used to evaluate the influence of the J-grooves on the vortex distribution in the conical section. The equation is expressed as

$$\frac{D\omega}{Dt} = (\omega \bullet \nabla)V - \omega(\nabla \bullet V) + \frac{\nabla \rho_m \times \nabla p}{\rho_m^2} + \nu(\nabla^2 \omega) \tag{12}$$

The left side of the equation represents the rate of vorticity change; the vorticity generation of the fluid is shown on the right side, which includes four parts: the first term $(\omega \cdot \nabla)V$ is the stretching term of the vortex, representing the stretching and torsion of the vortex. The second term $\omega(\nabla \cdot V)$ is the expansion term of the vortex caused by expansion and contraction. The third term $(\nabla \rho_m \times \nabla p)/\rho_m^2$ is the baroclinic torque term, which is zero when cavitation is not considered. The last term is the viscous diffusion term, which can be ignored.

The change of vorticity mainly occurs on the surface of the vortex belt. Because of the vortex flow and the back jet flow, there are two vortices with opposite values, one pair of stretch and compression, and one pair of expansion and contraction in the flow field, so only the vortex expansion term is analyzed in this paper. Because of the viscosity of the fluid, vortices will always be generated when interacting with the stationary wall. The vortices are transported from the place where the vortex is strong to the place where the strength is small, until the vortex intensity is equal everywhere. As shown in Figure 15, there is a positive and negative vortex in the flow field, which causes the vortex to rotate. The results show that the vortex pair is weakened and no longer has a spiral form with the increase in load. When the J-grooves are added in the conical section, the development trend of the tension–compression term and expansion–contraction term in the vorticity is suppressed, and the development of the vortex rope is restrained. Compared with the draft tube without J-grooves, the vorticity in the draft tube with J-grooves is smaller at the same working condition. The installation of the J-grooves exerts a positive effect on reducing the vortex generated by the axial flow.

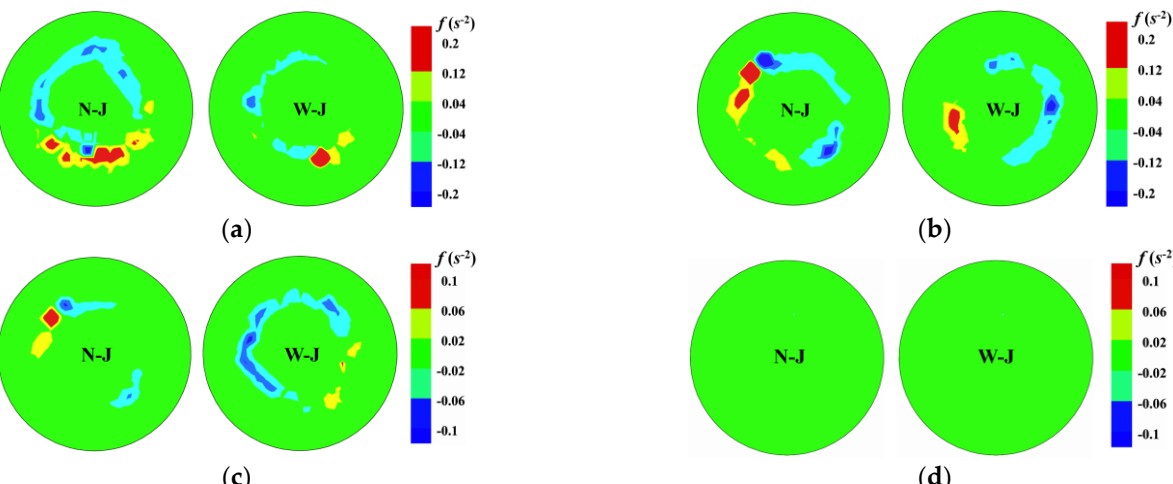

**Figure 15.** The transient distribution of vortex expansion $\omega(\nabla\cdot V)$ on the monitoring surface TP1. (**a**) $Q^* = 53\%$, (**b**) $Q^* = 69\%$, (**c**) $Q^* = 84\%$, (**d**) $Q^* = 100\%$.

The vortex suppression effect of J-grooves under partial load conditions is shown in Figure 16. In the part-load range, the water flow in the draft tube from beginning to end is composed of main water flow, backflow, and return flow [39]. When the turbine is operating at $Q^* = 53\%$ (DPL) conditions, the backflow and return flow in the draft tube are no longer symmetrically distributed, and part of the return flow forms a flow that separates between the main water flow and the backflow. The circumferential velocity of the flow is relatively large at DPL operating conditions; then, the return flow obtains sufficient rotation speed and energy. As seen in Figure 16a, under the action of the circumferential and axial velocities, an obvious spiral vortex appears. The vortex belt occupies the conical section and part of the elbow section with obvious eccentricity. After adding the J-grooves, the J-grooves disperse the return flow, which is out of the synchronous and concentric rotating row, reducing the vortex intensity and the circumferential component of the flow velocity. The length of the spiral vortex belt is reduced; it only appears in the conical section. The eccentricity of the vortex belt reduces, and no vortex belt exists below the TP1 cross section.

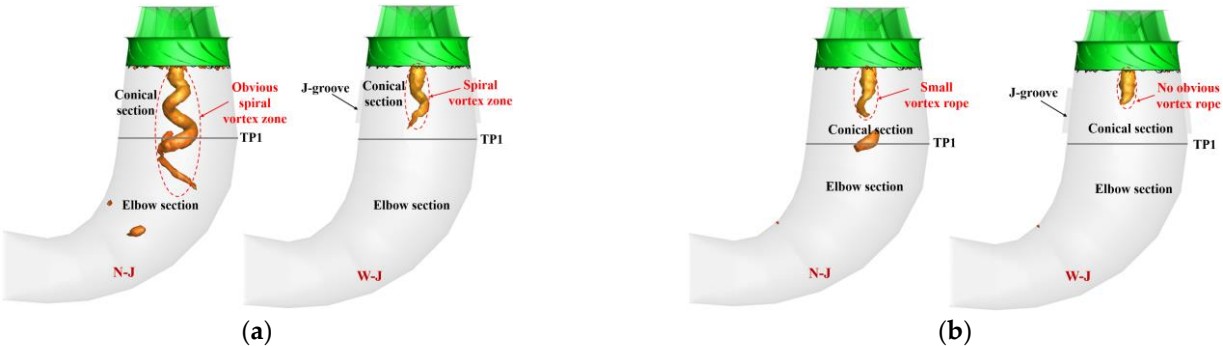

**Figure 16.** The vortex rope zone with or without J-grooves at $Q^* = 53\%$ and $Q^* = 69\%$ operating conditions. (**a**) $Q^* = 53\%$ (DPL), (**b**) $Q^* = 69\%$ DPL).

For $Q^* = 69\%$ (PL), due to the reduction in velocity circulation in the draft tube without J-grooves, the spiral degree of the vortex is lower than that of $Q^* = 53\%$, the length of the vortex rope is shorter, only a smaller vortex rope appears in the conical section. After the J-grooves are added, the circumferential velocity distribution reduces effectively. There is no obvious vortex rope in the draft tube; a shorter cylindrical vortex exists in the conical section, as shown in Figure 16b. The vortex belt only exists in the initial section of the straight cone without obvious eccentricity; therefore, J-grooves effectively reduce the vortex intensity and increase the pressure level of the vortex core in the draft tube of a

Francis turbine, and the expected effect of the J-grooves on vortex suppression and energy dissipation under partial load conditions is achieved.

## 5. Conclusions

For the first time, the entropy theory was used to study the energy loss analysis under the draft tube optimized by the J-grooves; the specific positions of the energy loss in the runner and conical section were obtained by using the entropy theory. Different from the traditional method of studying vortex suppression in the draft tube, in addition to comparing the pressure pulsation and frequency results of different working conditions with and without J-grooves, the results of swirl intensity $S_{num}$ and vortex expansion term $\omega(\nabla \cdot V)$ were also compared. The main conclusions can be drawn as follows:

a. At DPL or PL condition, the entropy production on the wall of the conical section with J-grooves is obviously smaller than that without J-grooves for the peripheral velocity component at the outlet of the runner is suppressed, the energy losses near the J-groove are reduced.

b. The J-grooves has an axial guiding effect, reducing the circumferential velocity of the water flow into the conical section, so the stronger the swirl intensity enters the draft tube, the higher the effect of J-grooves on vortex suppression because the circumferential velocity of the water flow is reduced after the water flows through the J-grooves.

c. The circumferential velocity component Vu and the angular momentum were reduced effectively after installing J-grooves. As a result, the swirl intensity $S_{num}$ and the peak of the pressure pulsation spectrum were significantly weakened.

d. The J-grooves slow down the development trend of the expansion items $\omega(\nabla \cdot V)$ in vorticity, and the development of the vortex band is restrained. As a result, the area of the vortex rope was significantly smaller than that without J-grooves at part-load conditions.

The J-grooves exert a positive effect on vortex suppression and energy dissipation. The results of this paper can be used as a research basis in the future for vortex suppression and energy dissipation in a draft tube. Next, how to combine the J-grooves with the drain cone with reasonable length to improve the operating stability of the hydro unit under PL and DPL operating conditions.

**Author Contributions:** Conceptualization, C.N. and Z.L.; methodology, C.N.; software, C.N.; validation, S.L., T.G. and S.G.; formal analysis, Z.L.; investigation, Z.L.; resources, S.G.; data curation, C.N.; writing—original draft preparation, C.N.; writing—review and editing, Z.L.; visualization, S.L.; supervision, T.G.; project administration, S.L.; funding acquisition, Z.L. All authors have read and agreed to the published version of the manuscript.

**Funding:** This work was financially supported by National Natural Science Foundation of China (NSFC) (Grants no. 52069010 and 51969009).

**Institutional Review Board Statement:** Not applicable.

**Informed Consent Statement:** Not applicable.

**Data Availability Statement:** The study did not report any data.

**Conflicts of Interest:** The authors declare no conflict of interest.

### Nomenclature

| | |
|---|---|
| BEP | Best efficiency point |
| NV | No-vortex condition |
| PL | Partial load condition |
| DPL | Deep partial load condition |
| N-J. | No J-groove |
| W-J. | With J-groove |
| $k$ | Turbulent kinetic energy ($\mathrm{m}^2/\mathrm{s}^2$) |
| $\varepsilon$ | Dissipation rate ($\mathrm{m}^2/\mathrm{s}^3$) |
| $y^+$ | Non-dimensional wall distance |
| $\eta$ | Overall turbine efficiency (%) |
| $r$ | The position from the origin in the $x$-direction (m) |
| $R$ | The radius of reference radius (m) |
| $S_{num}$ | Swirl number |
| $t$ | Time (s) |
| $L$ | Length of J-groove (mm) |
| $W$ | Width of J-groove (mm) |
| $D_1$ | Nominal diameter of the runner (m) |
| $n_{11.}$ | Rated unit speed (r/min) |
| $Q_{11.}$ | Rated unit flow ($\mathrm{m}^3/\mathrm{s}$) |
| $Q^*$ | Corresponding flow ratios |
| $C_r$ | Reduction coefficient |
| $V_z$ | Axial speeds (m/s) |
| $V_u$ | Circumferential speed (m/s) |
| $\dot{S}_D'''$ | Local entropy generation rate ($\mathrm{W}\cdot\mathrm{m}^{-3}\cdot\mathrm{K}^{-1}$) |
| $\dot{S}_{\overline{D}}'''$ | Direct entropy production rate ($\mathrm{W}\cdot\mathrm{m}^{-3}\cdot\mathrm{K}^{-1}$) |
| $\dot{S}_{D'}'''$ | Turbulent entropy production rate ($\mathrm{W}\cdot\mathrm{m}^{-3}\cdot\mathrm{K}^{-1}$) |

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
