# Peer review of "The Effect of J-Groove on Vortex Suppression and Energy Dissipation in a Draft Tube of Francis Turbine"

_energies, doi:10.3390/en15051707_

Round 1

Reviewer 1 Report

The author has focused on the Effect of J-Groove on Vortex Suppression and Energy Dissipation in a Draft Tube of Francis Turbine. It is valuable in engineering design. However, as a science research paper, further improvement is necessary. And the following points should be considered carefully.

Major Comments to the author:

  1. The introduction contains less description of what was delivered to the literature so far in the field of Francis turbine. The cited literature is thin and most of the cited literature is old. It should be enriched with more relevant and recent references, and arrange as per the required style of the journal. I would strongly suggest the author add some recent literature.
  2. The hexahedral meshes look very nice. However, I can only see the description regarding the y+ value without any proof. I would suggest the author add a y+ curve to support what you have said in the description. And be proud of, especially looking at the curvature of the elements and resulting hexahedral mesh. Reliable and persistent piece of work!
  3. I am not sure if I understand the boundary conditions. The boundary conditions should be provided in a significantly more clear way, maybe with a separate heading, and it should cover all the important boundary parameters. For instance, have a look at the boundary conditions in this paper (https://doi.org/10.1007/s13762-021-03619-1) in the literature, and provide the relevant information.
  4. The overall results discussion could be described as a piece of good scientific work. The only thing I would expand on is the consequence of that analysis. Namely, the physics were properly simulated and the relevant reason for energy dissipation and Vortex Suppression was diagnosed. However, discussion on potential improvement in order to diminish the problem was not conducted. Preferably, the conclusions section could be improved by such discussion or at least a short paragraph indicating further work.
  5. Is there any specific reason for choosing the rated speed (600rpm) for the measurements in your study? Why not any other higher value of rotational speed. Because according to my knowledge, some Francis turbines can work at higher rotational speed (e.g. Maximum 1000 rpm). Please mention in the manuscript, if any.

Minor comments to Author:

  1. Fig. 1, providing the description/labels pointing at particular components would be valuable for readers. Maybe some additional “refreshment”? What is the ladder for from the point of view of the scientific experiment?
  2. I would strongly suggest authors add Nomenclature at the end of the manuscript, which will explain the symbols, units, and abbreviations, in order to make it easier for the readers.
  3. What kind of gridding/Meshing technology is applied to handle the moving boundary?
  4. The conclusion is a little bit longer. It should be refined by keeping the most important points such as new findings or new methods.
  5. Please avoid repetitions in the paper. The first paragraph of the conclusion, abstract, and the last paragraph of the introduction is giving the same information. Please consider deleting it, where possible.
  6. The main structure parameters and size should be listed in this paper.
  7. The novelty or academic contribution to this field should be indicated clearly.

Author Response

Dear reviewer :

 Please see the attachment,

Reviewer 2 Report

In the manuscript, the authors have considered the vortex rope in the draft tube as the major contributors to pressure pulsation at partial load conditions which causes hydro unit to operate unstably. Based on the prototype Francis turbine HLA 551-LJ-43 in the laboratory, J-grooves are designed. It is observed that using J-grooves show better performance on vortex control and energy dissipation in the draft tube of a Francis turbine at partial load conditions.

The study in the manuscript is novel and interesting conclusions are recorded. But the present form the manuscript requires some minor revision for addressing the following observations.

  1. Abstract needs to be more precise and comprehensive, abbreviations need to be eliminated from the abstract.
  2. The manuscript should be revised with respect to English language; for example, at many place the sentences are started with “To” (see lines 46, 140, 245, 253, 377 etc.), that is grammatical not correct. “To” may be replaced by “In order to”.
  3. Similarly, on line 132 line started with “And” is not grammatical correct.
  4. On line 278 replace “While when” by “While for the case”.
  5. After Eqs. (1) and (2), please explain what are Sk, RƐ and SƐ.
  6. On line 111 replace “C, C and C” by “C, C and C
  7. Please explain why the entropy production caused by heat transfer is not considered (as mentioned on line 144).
  8. On line 165 replace “……eddy-viscous frequency, s-1” by “…….eddy-viscous frequency having units s-1”.
  9. On line 228, the line “The experimental results of the vortex rope change with different operating conditions” is unclear.
  10. In the manuscript r is the radial position, R is the radius of draft tube section, why the ratio r to R is negative as mentioned on the line 312.
  11. In Section 5, title “Conclusion” should be replaced by “Conclusions”

Author Response

Dear reviewer :

Round 2

Reviewer 1 Report

Overall Comments:

I appreciate the authors for their amendments if any. However, I'm not sure if the changes have been made to the manuscript or not because for the majority of questions, I am unable to find details/information regarding the changes neither in the manuscript nor in the response file. I strongly recommend the author to highlight the changes (using Track changes or highlighting with Red color) in the manuscript or provide the amended details in the response file.

Author Response

Dear editor:Please see the attachment

Round 3

Reviewer 1 Report

I agree with the amendments made by the author in the revised manuscript. The revised version of the paper is publishable in the prestigious journal Energies. 
However, for the revised version, I have one minor recommendation to make.

1. In the conclusion, in Line 514, the author has used "for the first time" in the description. Since the entropy production theory has been used in past by several studies for analyzing energy loss. Therefore, I don't think the use of above-mentioned phrase is appropriate. I would strongly recommend the author to either delete this or rephrase it in some other way.